# How much is a chef's touch worth? Affective, emotional and behavioural responses to food images: A multimodal study

**Pedro J. Rosa** [1,2]*, **Arlindo Madeira** [3,4], **Jorge Oliveira** [1], **Teresa Palrão** [5,6]

1 HEI-Lab: Digital Human-Environment Interaction Labs, Lusófona University, Lisbon, Portugal, 2 Instituto Superior Manuel Teixeira Gomes (ISMAT), Portimão, Portugal, 3 Higher School of Administration Sciences (ESCAD), IPLUSO, Lisbon, Portugal, 4 Centre for Tourism Research, Development and Innovation (CiTUR), Polytechnic of Leiria, Leiria, Portugal, 5 Instituto Superior de Lisboa e Vale do Tejo/ISCE, Odivelas, Portugal, 6 Estoril Higher Institute for Tourism and Hotel Studies (CiTUR), Estoril, Portugal

* pedro.rosa@ulusofona.pt

## Abstract

### Background

Food aesthetics influences affective dimensions (valence and arousal) and subsequent emotional and behavioural responses in images presented in more traditional form, almost rustic in some cases, to the signature dishes of haute cuisine. However, the visual impact of images of haute cuisine dishes on consumers' affective and emotional responses compared to traditional dishes is still understudied.

### Methods

We recorded electrodermal activity, ocular movements and self-report affect of 35 volunteers while they performed a picture viewing paradigm using images of haute cuisine food, traditional food, and non-food. Additionally, the moderating role of age was examined.

### Results

Our results showed that subjects had higher feelings of pleasure and arousal toward images of food (haute cuisine and traditional dishes) compared to non-food images. However, no difference in self-report affect, physiological and behavioural responses was found between haute cuisine and traditional dishes. Interestingly, a moderating effect of age was revealed, reporting that younger participants had greater feelings of pleasure and shorter eye-to-screen distance towards traditional food than haute cuisine.

### Conclusions

As a whole, our findings suggest that food aesthetics could at least partially affect consumers' affective and emotional responses. Interestingly, physiological responses to food pictures seemed to be relatively independent of approach/avoidance motivational states, supporting the assumption that traditional visual restaurant menus with attractive images might be insufficient for eliciting intense positive emotions. This study also contributes to

**Data Availability Statement:** Anonymized raw data, study materials, preprocessing scripts and statistical analyses that can be found here osf.io/5ts73 The complete list of stimuli used in this

experimental task is shown in the supplementary data (https://osf.io/f2tcm).

**Funding:** The funders had no role in study design, data collection and analysis, decision to publish, or preparation of the manuscript.

**Competing interests:** The authors have declared that no competing interests exist.

advancing the understanding of the role that age plays in emotional impact when images of haute cuisine dishes are presented to consumers.

## Introduction

The activity of consuming food enables individuals to fulfil their survival instincts and perpetuate their genetic legacy [1]. Consequently, when people come across food in various settings, their inclination to eat is stimulated, particularly when the food appeals to their senses, such as its visual, olfactory and gustatory sensory modalities [2]. The manner in which people come into contact with food, particularly how it is presented, can shape their preference.

Most food marketing capitalizes on innate human responses, enticing individuals with visually appealing representations of food. Advertisements, displays, and packaging present food-related images which appear to be delectable and ready to be eaten in order to increase the desire to eat, i.e., the appetite [3]. These visual depictions can be even more appealing when it comes to haute cuisine (HC) food. In fact, there are some HC dishes that are more impactful for the visual senses as they use complex gastronomical techniques (e.g., foams, froths and bubbles) to be more aesthetically attractive [4]. Thus, these beautiful depictions of food may impact not only individuals' evaluation, affective and emotional responses but also their later consumption patterns. Nevertheless, there are additional factors related to the presentation of food that can influence an individual's cognitions, emotions and actions. As the ageing process is accompanied by lower efficiency in visual sensory processing, little is known about how images of HC food, that is, high aesthetically attractive food, affects the perception and subsequent emotional and behavioural responses in older individuals.

To this end, this study examines how the aesthetics in food images affects affective and emotional responses toward the food and how age acts as a moderating variable. By examining how images with HC dishes alter affective, emotional and motivational processing, this study contributes theoretically to a better understanding of how biologically relevant and aesthetically attractive stimuli alter downstream responses and attitudes while promoting the comprehension of the approach-avoidance theory [5]. This theoretical understanding will support the development of images that can promote HC food choices, offering marketing opportunities for both restaurant industry stakeholders and the broader food sector, emphasizing the important effect of age in consumers' minds when exposed to diverse culinary styles and food presentations.

### Food as a biologically relevant stimulus

Food stimuli are thought to be biologically relevant stimuli, as they are essential for survival. Food serves as the primary source of energy for living organisms, providing the necessary calories and nutrients required for bodily functions, growth, and daily activities [2]. Therefore, it is important for our brain to efficiently recognize edible items in the environment [6]. Our evolutionary success is heavily dependent on our reactions to food-related stimuli. As we are visually oriented organisms, the visual sensory experience of food is a highly evolved aspect of an organism's interaction with its environment, which can guide us in selecting suitable food sources and avoiding spoiled or harmful foods [7, 8].

Whenever we see something potentially edible, affective and emotional responses are triggered, even when it is only portrayed visually in an image [9, 10]. This occurs because food is a biological motivator, automatically set in motion an organism's evolved aversive and appetitive motivational systems [5, 11]. These motivational systems in turn, orchestrate the processing of

relevant information, emotional responses, and subsequent behaviour. Moreover, any relevant stimulus, such as an image, has the potential to activate either the appetitive (approach) system, the aversive (defensive) system, or both concurrently [12, 13]. Thus, tendencies for appetitive (approach) and aversive (avoidance) behaviours in response to positive and negative stimuli, respectively, would thus enhance the adaptation of the organism to its environment [14]. Appetitive motivational circuits are thought to be linked in the organism to approach positively valenced (pleasant) stimuli, whereas a defensive motivational system would serve to trigger avoidance behaviour away from negative (unpleasant) stimuli [15].

Research has shown that both real food and food-related stimuli such as images can be positively perceived and automatically activate reward regions in the brain, independently of ratings of subjective hunger/food craving [15, 16]. Neuroelectrophysiological studies have shown similar results, demonstrating that sustained preferential processing of food-related stimuli is similar to stimuli of explicit emotional content in comparison to neutral stimuli [17, 18]. This evidence is also supported by autonomic peripheral measures (e.g., electrodermal activity [19, 20]), ocular measures (e.g. [21, 22]) and self-reporting instruments [23–26]. However, food-related stimuli are not created equally with regard to the appetitive responses they elicit. Previous research has shown that some perceptual characteristics (i.e., colour or shape) lead to automatic approach behavioural responses, with more positive attitudes towards the food presented and subsequent choice [2, 27].

## Haute cuisine images as aesthetically attractive stimuli

The presentation of food images in a laboratory context allows researchers to focus only on vision, as opposed to real food in a restaurant environment, where the other senses directly interfere, as well as other environmental factors [28]. The viewing of food-related images works as a preparatory or anticipatory step in food intake, without exposure to the visual appearance and aromas of a food or dish [29]. Food aesthetics can alter expectations regarding the flavour that we will experience on the palate [30]. Based on this, suggestive food images have gained emphasis in recent decades with the proliferation of television programmes, specialized magazines and more recently, social networks and blogs dedicated to gastronomy, in what has been conceptualized as food porn [31].

In fact, we are equally attracted by images of high-calorie food dishes, which include a large part of popular Portuguese traditional recipes [32], as well as exclusive and sophisticated haute cuisine (HC) dishes, in which there is a symbiosis between flavour and food aesthetics as an art form [33, 34]. In the case of traditional Mediterranean cuisine, the visual attraction is due to the quality of the products and their freshness, placed on the plate in an unpretentious way, respecting homemade recipes, categorized as comfort food [35]. Traditional food products and dishes from regional cuisines constitute an important legacy of European culture, identity, and heritage, common to the people of a country [36]. However, we are witnessing a movement away from the consumption of traditional European cuisine by new generations, to the detriment of food known as fast food [37]. Thus, the fact that the consumption of traditional dishes has been decreasing in recent decades among young European adults causes the effect of novelty and attractiveness in relation to the traditional cuisine. With regard to HC, it is a multidisciplinary concept which comprises technical and creative facets, leading to increasing public and scientific interest [38]. HC can be thought of as "ephemeral art", harmoniously combining flavours, colours, textures, and shapes of the different ingredients [39]. According to Mengual-Recuerda et al. [21], HC consists of a gastronomic modality in which more effort is applied in the elaboration of the dishes, playing, apart from the textures and temperatures of the food, with the five senses with the main goal of offering a unique experience to the

customer. Hence, when someone experiences HC dishes, he/she is exposed to a variety of unique affective and emotional experiences. Although chefs state that the most important thing in gastronomy is the taste, the visual appeal exerts a decisive influence on the taste of the food we eat [31, 40]. The visual stimuli that food suggests are, like aroma, a primary sensory cue, which allows us to predict what we will find in the mouth [41].

According to Hagtvedt and Patrick [42], the primary function of well-designed dishes, such as HC dishes, is aesthetic gratification because prettiness is associated with hedonic pleasure. Thus, prettier dishes elicit higher pleasure due to the activation of brain areas linked to reward [43]. The aesthetics of HC dishes are associated by most consumers with the artistic aspect of this contemporary culinary style, which can be adapted to any regional cuisine, in all parts of the world [34, 38]. If we accept that HC is a form of artistic expression, despite the fact that flavour is the most important thing, the importance of aesthetics in the kitchen is undeniable. So, it is very important that the chromatic combination works in terms of visual stimulus [28, 44]. Aesthetics works in a way as the factor of harmony between flavour and visual appreciation [33, 45]. When compared to other arts, chefs associate cooking with architecture because a unique dish, like a building, must be appealing to the eye [34, 38, 45]. In this analogy with architecture, flavour parallels the functionality of the building, that is, it is not enough to be beautiful, it must fulfil its function [34, 38]. At the same time, aesthetics works as the signature of the chef and his art, because it is possible to identify which school/culinary movement he is associated with, such as in painting or architecture [34, 46]. Therefore, some HC dishes include, for instance, molecular spherification of ingredients or the stacking of food to create a more impactful presentation so the meals/dishes can look more attractive [47]. In fact, HC dishes can lead to different affective and emotional experiences as they are more visually appealing (attractive and sophisticated design) compared to traditional dishes [46, 48], and subsequently impact on consumer behaviour.

## Food perception with age

Choosing a dish from a menu with pictures seems quite simple, but is a very complex behaviour that is influenced by interacting factors [49] and depends on sensory aspects such as appearance and colour [50, 51], socio-economic factors [52], psychological/motivational factors [53] and sociocultural factors [54, 55]. In fact, food choice is a dynamic phenomenon [56] which is linked not only to historical times but also to how we perceive and choose food over our lifetime, that is, how food preferences change as we age [18, 57–59].

It is known that the ageing process is accompanied by lower efficiency in sensory processing but pleasure in food remains with increasing age [60]. However, analysis of this information appears to be different in adults of different ages. Recent evidence has shown that older adults perceive raw food images as more arousing compared to young adults [61]. Perhaps this arousing affective response in older adults might be explained by nutritional beliefs and dispositions toward their weight and health concerns [62]. Also, young people have been found to prefer snack food images in comparison to older adults, which can be explained by a pronounced craving for unhealthy foods due to a decreased brain signal in brain areas linked to regulation processes of eating behaviour [47, 63]. While an adolescent or young adult reacts to stimuli of food which is dense in energy and calories in an impulsive and hedonic way, led mainly by the suggestion of taste that the image shows and reward-seeking behaviour [46], an older adult appears to take nutritional and health factors into account [45]. Yet, the effect of age on perception of images depicting food dishes has not been studied, either in dishes of HC presented as an art form, or traditional dishes, without pretensions to aesthetic harmony. While prior studies have explored the creative and innovative aspects of HC [45, 46, 64],

research on the effect of aesthetics of HC dishes on affective, emotional and behavioural responses is scarce in general. The theoretical framework of emotions in the present study is based on dimensional theories of emotion and approach-avoidance theory which postulate that emotional responses are founded on two affective systems—appetitive and defensive. These impact on information processing and activating approach-avoidance behaviours [65–69], but also autonomic responses such as skin conductance response to food images in people with eating disorders [70] and healthy controls to palatable food exposure [71].

As emotions are complex and multidimensional experiences, we considered it crucial to measure these phenomena through three response systems: cognitive (subjective affective states), physiological (objective autonomic responses) and behavioural (actions), while images of traditional and HC dishes are displayed. Accordingly, the current study fills this research gap by exploring how consumers respond in terms of affect, emotion and behaviour to images of HC dishes (well-designed, complex, and aesthetically attractive) versus images of traditional Portuguese cuisine dishes (careless design, simple and aesthetically less attractive) versus non-food images (control) using a multimodal approach and simultaneously examining the potential moderating effect of age. Based on previous literature, we hypothesize that: H1) Both food images (HC and traditional dishes) are perceived more positively and as arousing, eliciting larger autonomic responses and more approach behaviours than non-food images; H2) Images of HC dishes, due to their attractiveness, are perceived more positively and as arousing and trigger larger autonomic responses and more approach behaviours than images of Portuguese local dishes; and H3) Age moderates the relationship between the type of food dishes and affective/autonomic responses and approach-avoidance behaviours.

## Method

### Sample

The sample was recruited in a university campus in Lisbon for participation in a study related to gastronomy. Power analysis with sjstats package version 0.18.2 [72] showed that a sample of at least 26 subjects provided a power of 80% ($p < 0.05$) to detect a medium effect size ($d = 0.5$) assuming an ICC of .05. Therefore, the sample comprised 35 Portuguese adults aged between 18 and 70 years with an average age of 31.4 years (SD = 16.1 years), with 22 women and 13 men. The sampling was non-probabilistic (convenience sampling), consisting of students of Lusófona University in Lisbon, Portugal. Regarding demographic features, most were single (n = 26), but three were married and five were divorced. As for education, most had secondary education (n = 28), three had a master's degree and one had a PhD. Therefore, most of the sample were students (n = 18), eight were active workers and three were unemployed. Only three participants were from a rural region, whereas most were from urban regions. As regards religion, most were Catholic (n = 14), ten were atheist, five were agnostic, two were from an evangelical church and one participant did not respond. Only healthy adults with no history of psychiatric or neurological pathology were included in the study.

### Experimental task

An active viewing task consisted of the presentation of 54 pictures (18 pictures showing haute cuisine food, 18 pictures showing traditional food provided by José Avillez, a two-Michelin-starred Portuguese chef, along with 18 neutral pictures retrieved from the International Affective Picture System—IAPS [73] describing objects. The complete list of stimuli used in this experimental task is shown in the supplementary data (https://osf.io/f2tcm).

The experience took place at 11:00 am, to coincide with the period that lies between breakfast and lunch in Portugal. As a rule, the Portuguese have breakfast at 9:00 am and lunch at

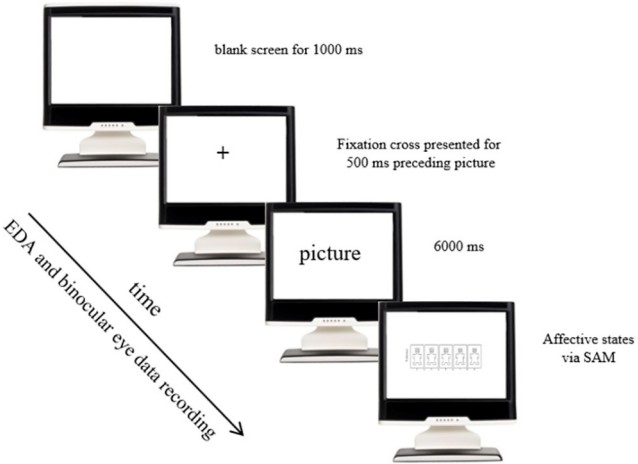

**Fig 1. Design of the active viewing task.**

1:00 pm. Considering that all participants were of Portuguese nationality, we can conclude that there are similar food consumption patterns, since Portuguese cuisine is part of the Mediterranean diet. Care was also taken to exclude vegetarian participants, considering that the images chosen contain animal protein. The criteria for choosing the traditional food stimuli took into account representative dishes of traditional Portuguese recipes, known by all Portuguese people. This recognition was verified with the exhibition of these images in a pre-test with Portuguese students of different ages. We opted to choose a diversity of images of dishes with animal protein, from sea and land. Regarding the choice of HC dishes, images of 35 dishes were made available by the chef José Avillez and a selection of 18 of these was made, considering their diversity, as was done with the selection of traditional cuisine dishes (fish, seafood, and meat). The most praised images in a pre-test carried out with Portuguese students of different ages were also taken into consideration. We also considered the choice of dishes with different colours in the stimuli of the two types of cuisine.

The authors of this study have permission to use the pictures for this research. Each picture was presented for 6000 milliseconds preceded by a blank screen for 1000 milliseconds followed by a central fixation dot for 500 milliseconds before each picture. All pictures were randomly presented in a 1280x1024-pixel resolution screen. All pictures were followed by the Self-Assessment Manikin (SAM) as depicted in Fig 1. This method was based on an active viewing task, in which the participants were required to evaluate the images on two predefined scales for assessing the affective/emotional reaction to each image, in a laboratory setting that was free of factors that could influence the emotional reaction to food images. The SAM was presented on the screen until a computer keyboard response (a number between 1 and 9) was collected from the participants.

### Measures

**Affective impact.** Subjective affective responses to the presented images were assessed via the Self-Assessment Manikin (SAM; [74]). The SAM is a pictorial measure that assesses affective responses to pictures in the dimensions of pleasure, arousal and dominance. However, the dimension of dominance was not evaluated in this study as it is positively correlated with the pleasure dimension [75, 76]. The SAM affective dimensions are both rated on a nine-point scale as shown in Fig 2.

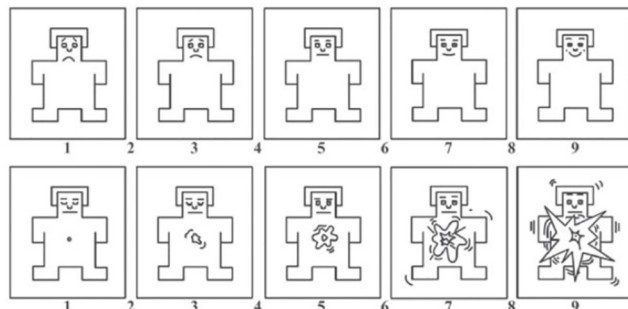

**Fig 2. Self-Assessment Manikin.** In the top panel, the left-most point of the SAM on the pleasure dimension denotes being very unhappy or unsatisfied, while the right-most SAM point reflects being extremely happy and pleased. In the bottom panel, the left-most picture is selected to show relaxed and calm, whilst the right-most SAM is related to being highly aroused due to extreme excitement.

**Physiological and behavioural activity.** *Skin conductance response.* It has been shown that emotional content increases autonomic activity, reflected in different physiological responses [77, 78]. Electrodermal activity (EDA) is the umbrella term that refers to autonomic changes in the electrical properties of the skin [79]. The EDA includes both background tonic (skin conductance level: SCL) and rapid phasic components (skin conductance responses: SCRs), which are manifestations of sympathetic activity. During affective elicitation via pictures, SCR amplitude increases with the subjective assessment of the emotional arousal of the stimulus, regardless of valence [80].

*Eye-to-screen distance (ESD) as a measure of motivated behaviour.* A large body of literature provides support for the idea that the evaluation of a stimulus (e.g., a picture) is linked to the tendency to move toward it when a positive evaluation is made (e.g., a delicious meal) or to move away from it whenever a negative evaluation is made (e.g., a meal you dislike) [81]. As this affective evaluation is automatic [75], behavioural predispositions toward the stimulus can be directly measured through body posture, indicating an approach behaviour whenever the distance between the participant and the stimulus decreases or an avoidance behaviour when the distance between the participant and the stimulus increases [82]. Along with a wide range of well-known ocular metrics (see [83, 84]), eye tracking systems can also provide an easy and intuitive eye measure linked to body posture [85]–the eye-to-screen distance (ESD). This measure corresponds to the mean distance from both eyes to the eye tracker screen (gaze vector length) as shown in Fig 3.

## Procedure

Upon arrival at the Laboratory of Experimental Psychology, participants were seated in a chair in a sound-proof cubicle with constant illumination (42 lux). Before the beginning of the experiment, each participant read and agreed with the informed consent form. The objective explained to participants was that participation was important to understand the reactions to different types of food. This study was approved by the Ethics Committee for Scientific Research of the School of Psychology and Life Sciences of Lusófona University in Lisbon, where this study was carried out. After agreeing with the informed consent, the participants completed a brief questionnaire to collect sociodemographic data. After completing the socio-demographic form, the participants were connected to the biosignalsplux (Plux Wireless Bio-Signals SA) system for psychophysiological recording, consisting of two electrodes for electrodermal activity (EDA).

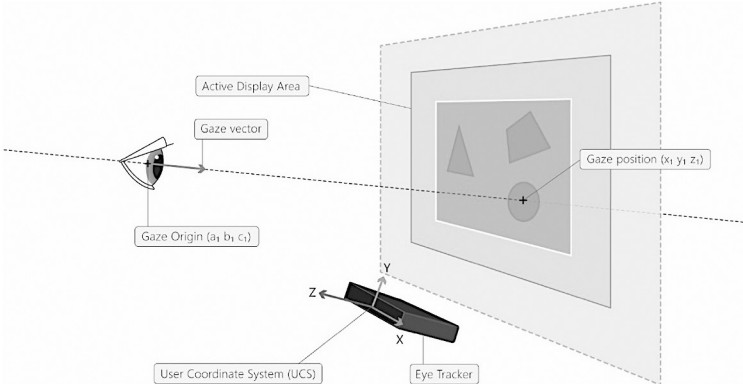

**Fig 3. The eye-to-screen distance.** The ESD is calculated based on the length of the vector from the centre of each eye to the user coordinate system (UCS) origin point on the eye tracker.

The biosignalsplux explorer is a four-channel, analogue, wearable, body-sensing platform which can record a wide range of physiological signals and transmit them via Bluetooth. EDA was measured using 11 mm silver/silver chloride (Ag/AgCl) pre-gelled and self-adhesive disposable electrodes, placed on the distal phalanges of the index and middle fingers of the non-dominant hand [72]. It should be noted that the participants received instructions to place the electrodes by themselves. The signal quality was always checked visually on the screen of the PC before starting the experimental task. Electrode placement was adjusted if any signal was of poor quality. In this study we recorded EDA via Opensignals software at a sampling rate of 1000 Hz. Then, the participants were seated in front of the Tobii T60 eye tracker for eye tracking and stimuli presentation. The active viewing task was designed in Tobii Studio (version 3.2.1), which is the stimulus presentation software from Tobii eye trackers. Binocular eye data was recorded with a temporal resolution of 60 Hz. The eye tracker calibration was performed using a nine-point calibration grid at the start of the task [86, 87]. The eye tracker was used to both ascertain eye movements during image presentation and to compute the ESD which is associated with motivated approach-avoidance behaviour. The participant's response to the SAM scale was provided through the keyboard. In this study we used a fast light meter in order to synchronize the onset of pictures displayed on the eye tracker screen while recording the EDA signal [88]. The light sensor was placed at the top on the right side of the eye tracker and generated a synchronization marker whenever the 50x50px square changed from white to black. According to the information in the informed consent form, the participants were informed that they would be exposed to food images showing traditional dishes. At the end of the task, the experimenter provided a debriefing explaining the general objectives of the study. The participants disconnected the electrodes and left the room at the end of the study. The duration of the data collection procedure was approximately 25 minutes. The authors had no access to information that could identify individual participants during or after data collection.

**Data reduction and statistical analysis.** Firstly, a missing value analysis was conducted. The percentage of missing values ranged from 2.9% (n = 1) for affective ratings to 20% (n = 7) for ESD. No missing values were found for SCRs. Regarding sociodemographic variables, only one missing value (2.9%) was found for religion. As missing data represented fewer than 5% for all study variables and were completely random (Little's MCAR test with p > .05), no imputation was made.

Raw EDA data were down-sampled at 200 Hz time resolution, filtered with a low-pass second-order Butterworth bandpass filter with a cut-off frequency at 5 Hz. SCRs were detected automatically via AcqKnowledge 4.1 software routine assuming three criteria: (a) 1–4 s temporal window after each picture onset; (b) default minimal amplitude for SCR detection > 0.3 µs; and c) a rejection threshold of 10% based on the participant's largest SCR peak [89, 90]. The amplitude of the SCR for each trial was computed and then averaged for each picture category. The SCR amplitude for each trial was range-corrected following the recommendations of Lykken et al. [91] and square root-transformed to meet parametric test assumptions [84].

Regarding eye data, the raw signal from both eyes was inspected and cleaned using the Tobii T60 tracker's validity score, which indicates the confidence level that the left and right eye have been correctly identified. The values range from 0 (high confidence) to 4 (eye not found). Validity scores of 4 in both eyes, which usually happen during blinks, were removed, leaving approximately 80% of total data per eye. ESD was calculated based on the estimated vector length (mm) from both eyes to the eye tracker screen during the picture presentation and corrected for the baseline (mean ESD 6s/mean ESD -1 s). Distances lower than 300 mm or above 900 mm (± half of ESD during calibration, that is 600 mm) were excluded (0%; n = 0). The corrected mean ESD was computed for each picture category.

As regards statistical analyses, the active viewing task was designed such that two planned contrasts could discriminate between specific emotional processing to food pictures. (1) neutral versus emotional pictures: neutral/non-appetitive stimuli (control) contrasted with emotional/appetitive stimuli (traditional + haute cuisine food); (2) traditional food versus haute cuisine: emotional pictures of traditional dishes contrasted with emotional pictures of haute cuisine food. A linear mixed model (LMM) was run for each dependent variable (four in total): two for affective ratings (pleasure and arousal dimensions); one for EDA (SCR amplitude); and one for eye data (ESD). We considered two fixed effects (picture category and age) and two random effects (participant and picture). Age was included as a continuous variable in the model. The LMMs were fitted using restricted maximum likelihood (REML) and the degrees of freedom were Satterthwaite approximations [92]. Simple effects were tested at one standard deviation above and below the mean [93]. We computed Cohen's d based on estimated marginal means. Planned contrasts were performed using Tukey's method [94]. All statistical analyses were performed using JASP software version 0.16.3 (JASP Team, University of Amsterdam) for Windows. The significance threshold was set to 5% in all analyses.

## Results

### Descriptive statistics

Table 1 below summarizes the descriptive statistics for the dependent variables.

**Table 1. Mean and standard deviation values for each dependent variable as a function of image category.**

| Dependent variables | Neutral | | Traditional | | Haute | |
| --- | --- | --- | --- | --- | --- | --- |
| | pictures | | Food | | Cuisine | |
| | M | SD | M | SD | M | SD |
| Pleasure | 4.04 | 2.14 | 5.19 | 2.37 | 4.80 | 2.35 |
| Arousal | 3.60 | 2.33 | 4.51 | 2.56 | 4.30 | 2.46 |
| SCR amplitude * | 0.05 | 0.21 | 0.05 | 0.22 | 0.08 | 0.36 |
| ESD* | 1.00 | 0.04 | 1.00 | 0.05 | 1.00 | 0.04 |

• Baseline corrected

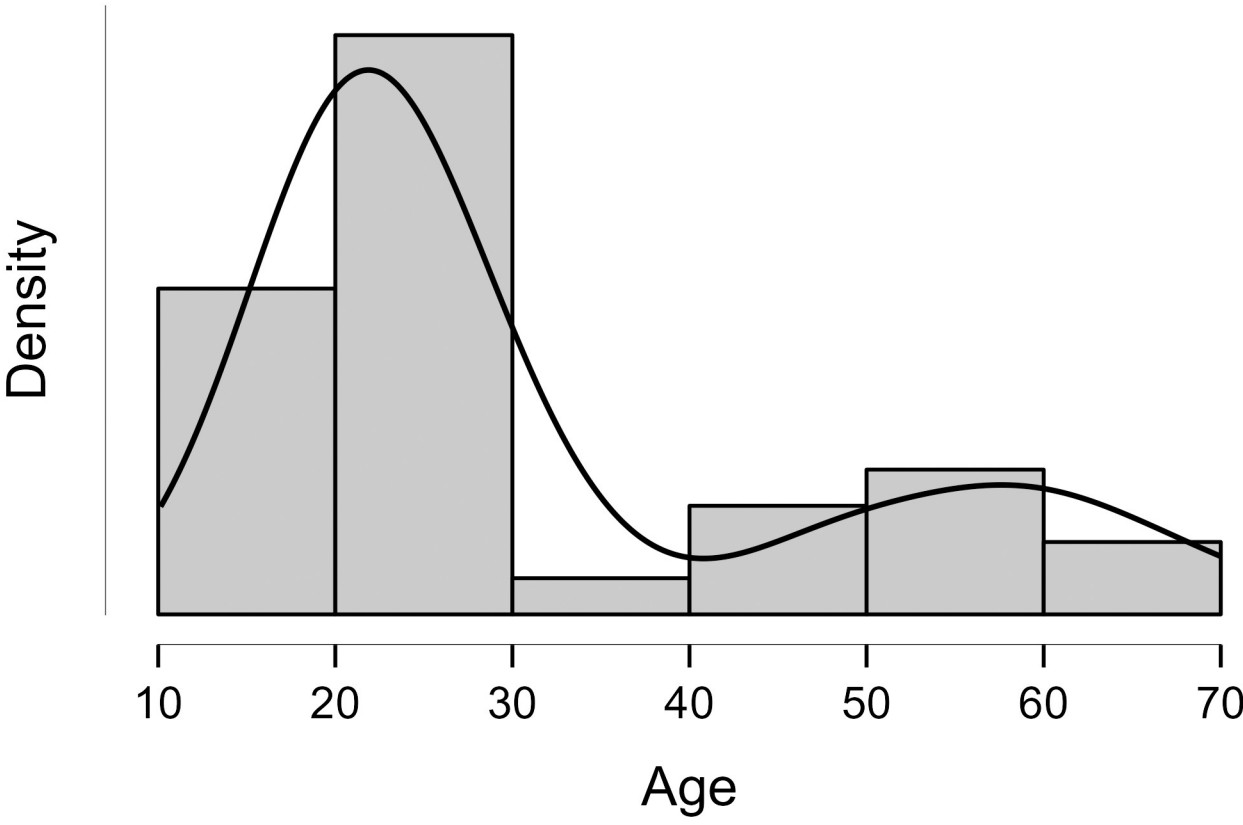

**Fig 4. Histogram showing the age distribution of study participants (*N* = 35).** Note: bin width based on Sturges' rule.

As age is a continuous variable, it is depicted on a histogram. As seen in Fig 4, the distribution is skewed; however, the values for skewness ($g_1$ = 1.23) and kurtosis ($g_2$ = -0.03) were between -2 and +2 which are considered acceptable in order to assume normal univariate distribution [95]. Any age above +3 SD or below -3 SD was considered an outlier. Based on this rule, no outliers were identified and removed.

### Affective ratings

For valence, the LMM indicated a significant interaction between picture category and age F (2, 1782.13) = 5.79, p = .003. A main significant effect of picture category was also found F(2, 277.27) = 18.50, p < .001, but not for age F(1, 32.84) = 4.02, p = .053. Planned contrasts for simple effects revealed that all participants rated food pictures (traditional + haute cuisine) as more pleasant than neutral pictures; however, this effect decreases with age, as shown by Cohen's d values. When contrasting traditional cuisine with haute cuisine, only younger participants rated traditional dishes (M = 5.14; SE = 2.44) as more pleasant than chef-prepared meals (M = 4.43; SE = 2.45) [t (90.36) = 3.02, p = .020, d = 0.63].

For arousal, the LMM revealed a significant main effect of picture category F(2, 273.90) = 12.15, p < .001. Neither a main significant effect of age F(1, 33.01) = 3.26, p =. 080 nor an interaction effect of picture category and age F(2, 1782.84) = 2.80 p = .061 was found. Planned contrasts showed that all participants rated food pictures (traditional + haute cuisine) as more arousing than neutral pictures, regardless of age. However, simple effects revealed no significant differences between contrasts across age levels (all ps > 0.05) as shown in Table 2.

**Table 2. Planned contrasts of mean affective ratings for total sample and across age levels.**

| Affective ratings | Age | Neutral pictures vs Food pictures | | | Traditional vs. Haute cuisine | | |
|---|---|---|---|---|---|---|---|
| | | $t$ | $p^t$ | $d$ | $T$ | $p^t$ | $D$ |
| Pleasantness | All | -5.43 | < .001 | **1.52** | -1.93 | .113 | 0.54 |
| | -1SD | -5.751 | < .001 | **1.20** | **3.02** | **.020** | **0.63** |
| | M | -5.43 | < .001 | **1.52** | -1.94 | .302 | 0.54 |
| | +1SD | -3.65 | **.003** | **0.72** | -0.34 | >.999 | 0.00 |
| Arousal | All | -4.78 | < .001 | **1.33** | -1.09 | .483 | 0.30 |
| | -1SD | -5.17 | < .001 | **1.08** | 1.52 | .570 | 0.32 |
| | M | -4.78 | < .001 | **1.34** | -1.09 | .862 | 0.30 |
| | +1SD | -3.12 | **.015** | **0.65** | -0.37 | .999 | 0.00 |

$^t$: adjusted p-values using the Tukey method. Bold values denote statistical significance at the p < 0.05 level.

## Physiological emotional response

As concerns SCR amplitude, results showed neither significant main effects [$F_{\text{image category}}$ (2, 1851.00) = 2.14, p = .118 and $F_{\text{age}}$ (1, 33.00) = 0.40, p = .843] nor a significant effect of interaction (2, 1851.00) = 0.77, p = .462. Planned contrasts showed no statistically significant differences (all ps > 0.05) as seen in Table 3.

## Approach-avoidance behaviour

As shown in Table 4, results indicated significant interaction between picture category and age on ESD F(2, 1538.169) = 4.99, p = .007. A main significant effect of picture category was also found F(2,1540.12) = 5.35, p = .005, but not for age F(1, 21.30) = 0.95, p = .340. When

**Table 3. Planned contrasts of mean SCR amplitude for total sample and across age levels.**

| Physiological measure | Age | Neutral pictures vs Food pictures | | | Traditional vs Haute cuisine | | |
|---|---|---|---|---|---|---|---|
| | | $t$ | $p^t$ | $d$ | $T$ | $p^t$ | $d$ |
| SCR amplitude | All | -1.31 | .346 | 0.01 | 2.12 | .068 | 0.10 |
| | -1SD | -0.72 | .978 | 0.00 | -1.01 | .895 | 0.01 |
| | M | -0.54 | .996 | 0.00 | -1.34 | .710 | 0.01 |
| | +1SD | -0.04 | >.999 | 0.00 | -0.90 | .938 | 0.00 |

$^t$: adjusted p-values using the Tukey method

**Table 4. Planned contrasts of mean ESD for total sample and across age levels.**

| Behavioural Measure | Age | Neutral pictures vs Food pictures | | | Traditional vs Haute cuisine | | |
|---|---|---|---|---|---|---|---|
| | | $t$ | $p^t$ | $d$ | $T$ | $p^t$ | $d$ |
| ESD | All | 0.19 | .977 | 0.01 | 0.99 | .544 | 0.01 |
| | -1SD | -0.21 | >.999 | 0.01 | **2.91** | **.022** | **0.14** |
| | M | 0.19 | >.999 | 0.01 | -0.99 | .905 | 0.01 |
| | +1SD | 0.48 | .997 | 0.02 | 1.51 | .567 | 0.07 |

$^t$: adjusted p-values using the Tukey method. Bold values denote statistical significance at the p < 0.05 level.

contrasting traditional cuisine with haute cuisine, only younger participants showed a short ESD for traditional dishes (M = 0.998; SE = 0.003) in comparison to chef-prepared dishes (M = 1.000; SE = 0.003) [t (1541.02) = 2.91, p = .020, d = 0.15]. No other significant contrasts were found (all ps > 0.05).

## Discussion

The present study was designed to examine whether dishes from a two-Michelin-starred chef have a different impact on affective dimensions (valence and arousal) and subsequent emotional and behavioural responses in comparison to traditional dishes, also considering a potential interaction with age.

Our results showed that both images of HC and traditional dishes were perceived as more pleasant and arousing than non-food images. This seem to be consistent with the findings of several studies that reveal that food cues can preferentially engage attention over non-food cues [96–99]. In addition, some food images cause an attraction that can be classified as sensual or even pornographic [100]. However, no statistical differences in physiological (SCR amplitude) and behavioural responses were found between food and non-food images, partially confirming our first hypothesis (H1). This might be explained by the fact that no very contrasting images in terms of hedonic valence were presented, (i.e., positive vs neutral). Some studies have shown larger SCR responses to unpleasant food images (e.g., worms/insects) in comparison to neutral and pleasant food images [101, 102]. Unpleasant food images can elicit aversive or defensive responses, whereas pleasant images, as in our study, can elicit appetitive responses [101]. This suggests that electrodermal activity may be more useful and give more insights in experimental tasks with very contrasting exposures, compared to our experimental task. Furthermore, there was a relatively high percentage of participants with few SCRs or no SCRs at all during exposure to food images. This could indicate that neither HC nor traditional dishes were arousing enough to trigger SCRs, which is corroborated by the arousal mean values for each type of dishes that were lower than scale middle point (i.e., 5 on a nine-point scale).

Concordantly, ESD also showed no difference between food images (HC and traditional) and non-food images. As can be seen by the pleasantness mean values, food images were not perceived to be very pleasant (i.e., around 5 on a nine-point scale), which might have led to a weak approach behaviour. It is also possible that a positivity offset effect might have emerged, reflecting the tendency of the appetitive motivational system to respond more strongly in the absence of or weak emotional input [103, 104]. Based on this, participants might have responded more positively to neutral images (non-food images), triggering a weak approach behaviour and, therefore, minimizing the differences in ESD.

In relation to the differences between HC dishes and traditional dishes, hypothesis H2 was not confirmed. We formulated our initial hypothesis on the assumption that attractiveness of the HC food would contribute to the differences in affective, physiological and behavioural responses compared to traditional dishes [42, 52, 105]. Overall, participants reported similar feelings towards HC and traditional dishes. This suggests that the aesthetics do not seem to explain our results, as our participants respond similarly in terms of emotion and motivation toward both types of dishes. This null effect might be explained due to familiarity and cultural identity to traditional Portuguese dishes. It is possible that participants feel that such traditional dishes are preferable and are perceived as more positive and more arousing [106]. This assumption is supported by some studies that have shown that consumers tend to evaluate their traditional food more favourably than non-traditional food [107, 108]. Furthermore, the participants disposition to avoid novel or non-traditional food did should be taken into

account. Indeed, participants with a higher avoidance reaction to artistic dishes with novel design are more likely to make favourable evaluations of traditional food, with higher feelings of pleasure and arousal and subsequent approach responses towards them [109].

The hypothesis of the potential moderating role of age (H3) was confirmed. Our results revealed that traditional dishes were perceived as more pleasant compared to exclusive chef-prepared dishes, but only in younger participants. The interest of younger people in what is called comfort food can be justified by the novelty effect, when compared to older individuals. In other words, a gradual departure from the identity patterns of regional gastronomy makes images of traditional food exert a novelty effect on millennials [38]. On the other hand, it may be stated that traditional foods bring nostalgic memories (not always positive) to older participants, related, for example, to their past [110]. For middle-aged participants (M = 37.5), traditional food images had lower ESD (greater pleasantness) than images of chef-prepared food.

Regarding participants from other ages (younger and older), there were no significant differences between neutral versus food images (chef and traditional). Since traditional food dishes are high in calories, this result can be explained by the fact that middle-aged people are more concerned about health care, which makes higher-calorie food less appealing [111]. On the other hand, the greater sensitivity of middle-aged individuals to food as an art form may explain the greater impact of images of dishes created exclusively by chefs, where the aesthetic design has a decisive weight [43, 99]. That is, age, culture and lifestyle can explain the motivational attitude and stimuli towards food [112].

## Limitations and future research

The study was conducted in a lab environment instead of in a real restaurant context, where food and non-food images were presented. This decision made it possible to isolate vision from the other senses, allowing the participants to focus only on the visual stimuli provoked by the dish images, something that is not possible when we taste food. In addition, it prevented participants from being distracted by other components that are part of the gastronomic experience in a restaurant environment (e.g., music, service, decoration, customers). On the other hand, although food images are a widely used tool in studies related to research on emotional stimuli, they differ from real dishes in terms of colour, intensity, and real background context. Future studies should use more ecological sensory stimuli, such as the visualization of actual HC and traditional dishes, exposing participants to other types of stimulation, such as olfactory and gustatory. Another limitation was the lack of knowledge of the level of familiarity and neophobia of the participants with HC. These variables would have helped to explain the inconsistency of results between subjective, physiological and behavioural measures.

Also, the lack of control of hunger level of the participants could be considered a limitation, even though they were assessed in the morning, a few hours before lunch time. It is known that fasting individuals attribute relatively more pleasure and arousal to large, caloric doses of food, which is consistent with the fact that eating in a state of hunger leads to a conditioned increase in the judgments of the pleasantness of high-calorie food, which is the case of Portuguese traditional dishes [113].

The small sample size is also a shortcoming of this study. The current study was limited to 35 participants, mostly students, thus impairing the external validity of our experimental task. The small sample size is also a potential explanation for some null results a larger sample with more statistical power may uncover significant effects, namely when it comes to interaction (moderation). Futures studies should gather a full age spectrum sample, for example ranging from children to older adults, allowing the effects of age in a wider range under study. Nevertheless, because the HC dishes might elicit a distinct pattern of responses, it would be relevant

to replicate and extend the present study to other countries. In further studies, it would also be appropriate to include other unobtrusive measures acquired by means of a high-definition camera that can be used for facial emotion recognition (e.g., NOLDUS FaceReader software) or cardiac activity analysis via remote photoplethysmography.

## Conclusions

The present study aimed to gain insights into consumers' affective, emotional and behavioural responses to images of dishes from a two-Michelin-starred chef. We approached this question by exposing them to images of HC dishes, traditional dishes and non-food while gathering distinct measures of emotions. Based on dimensional theories of emotion, we considered both valence and arousal through self-reporting, physiological and behaviour indicators [114, 115]. Our findings suggest that traditional dishes seem to stimulate a more pleasant affective response in younger participants, with more pronounced motivated approach behaviour when compared to HC dishes, although physiological and behaviour measures yielded different results. The findings reenforce the importance of measuring emotions through several levels of response, in addition to collecting participants' subjective feelings, given their active role in triggering consumer behaviour [116]. The literature has shown that subjective cognitive evaluations and physiological responses can sometimes be incongruent [99, 117, 118]. Accordingly, from our perspective, ESD might be a more sensitive measure to capture valence and arousal effects of food images. This study provides some important contributions to the discussion of gastronomic and consumer neuroscience. From the practical point of view, food businesses should rethink aesthetics as the most important factor regarding emotion as our findings suggest that images of traditional dishes can elicit similar emotional arousal and pleasure compared to images of HC dishes, despite the careless design and unattractiveness. Our results also support the assumption that traditional restaurants' visual menus with attractive images might be insufficient for eliciting positive, intense emotions; therefore, HC restaurants can create multi-sensory digital menus (e.g., video + audio or even smell) to promote the positive emotional impact of HC cuisine. Furthermore, the study contributes to advancing the understanding of the importance of age in emotional impact when food images with different presentations are displayed. The current research offers marketing opportunities for the players involved in the restaurant business, as well as the food industry in general, by bringing to the discussion the importance of age in consumers' brains when they are bombarded with disparate culinary styles and presentations, promoting the future use of age-based customized restaurant menus.

## Acknowledgments

Thanks for data collection and data preprocessing are owed to Jessica Carvalho. We also thank students of Lusófona University for lending their eyes and skin to our laboratory recording equipment.

## Author Contributions

**Conceptualization:** Pedro J. Rosa, Arlindo Madeira, Jorge Oliveira, Teresa Palrão.

**Data curation:** Pedro J. Rosa, Arlindo Madeira, Jorge Oliveira.

**Formal analysis:** Pedro J. Rosa.

**Investigation:** Pedro J. Rosa.

**Methodology:** Pedro J. Rosa, Jorge Oliveira.

**Writing – original draft:** Pedro J. Rosa, Arlindo Madeira, Jorge Oliveira.

**Writing – review & editing:** Pedro J. Rosa, Arlindo Madeira, Jorge Oliveira, Teresa Palrão.

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
