## [Decision Letter · Decision Letter 0]

2 Jul 2023

PONE-D-23-10020How much is a Chef’s touch worth? Affective and emotional responses to food images: a multimodal studyPLOS ONE

Dear Dr. Rosa,

Thank you for submitting your manuscript to PLOS ONE. After careful consideration, we feel that it has merit but does not fully meet PLOS ONE’s publication criteria as it currently stands. Therefore, we invite you to submit a revised version of the manuscript that addresses the points raised during the review process.

We look forward to receiving your revised manuscript.

Kind regards,

Alhamzah F. Abbas, PhD student

Academic Editor

PLOS ONE

Journal Requirements:

“This study was funded by the (BLINDED FOR REVIEW PURPOSES).”

3. We note you have included a table to which you do not refer in the text of your manuscript. Please ensure that you refer to Tables 3 and 4 in your text; if accepted, production will need this reference to link the reader to the Table.

Reviewers' comments:

Reviewer's Responses to Questions

**Comments to the Author**

1. Is the manuscript technically sound, and do the data support the conclusions?

Reviewer #1: No

Reviewer #2: No

2. Has the statistical analysis been performed appropriately and rigorously? 

Reviewer #1: No

Reviewer #2: Yes

3. Have the authors made all data underlying the findings in their manuscript fully available?

Reviewer #1: No

Reviewer #2: Yes

4. Is the manuscript presented in an intelligible fashion and written in standard English?

Reviewer #1: No

Reviewer #2: Yes

5. Review Comments to the Author

Reviewer #1: In this study the authors investigate subjective and physiological responses to food images and non-food control stimuli. Valence and arousal ratings, electrodermal activity and eye-tracking responses were collected from 35 Portuguese participants. Age-related effects were also examined.

The main advantages of the study are (1) its levels of investigation (subjective and physiological); (2) interesting research question.

Although the paper has several strengths, I do not recommend it for acceptance in its current form. Please, find some critics below.

ABSTRACT

The theoretical background of the study is unclear in the abstract as there is a mismatch between the content in the Background paragraph and in the Conclusions paragraph.

What does the eye-to-screen distance exactly mean (as a variable)? I’m not sure if it is the correct term to use.

INTRODUCTION

The theoretical background is too diverse. The focus point is not clear enough. Why is it important to conduct a multimodal study on affective responses to food images? What is the gap exactly that the authors fill in with their study?

METHODS

The sample was recruited based on convenience sampling, but the relatively low number of participants and the homogenity of the sample may be a limitation to generalize the results. I strongly encourage the authors to conduct a post hoc sample size calculation and a power analysis.

Page 8, line 170: The “civil status” of participants should be changed to demographic features, or to use the term sociodemographic variable consequently (see p 11, l 255).

P 8, ls 168-169: According to the text “ … 35 Portuguese students aged between 18 and 70 years....” . Maybe it should be corrected as the authors earlier mention that “students and faculty” participated in the study (l 166). Did the participants have prior psychological knowledge regarding the current study, or were they ‘naive’?

It is a question whether the actual motivational state, namely the hunger of participants or specific eating habits or diets could have an effect on their affective responses. Is there any information about the time of last food consumption, and type of it (e.g., main dish, snack, etc.), as well as specific diets and eating habits?

Please, clarify some details about the stimulus material: (1) Picture ID numbers from the IAPS would be really informative; (2) please, define the difference between the two food pictures or provide an example of each; (3) what was seen on the neutral pictures (e.g., geometric figures, natural scenes, neutral faces?)

If I understand it correctly, participants have placed the electrodes to themselves. It could cause measurement error, couldn’t it? Did the authors check it?

P 11, l 239: I am afraid, the ESD is not the optimal solution here. It is unclear why (and how) the ESD is informative about approach-avoidant motivation if the head is fixed during eye-tracking registration? The cited papers (lines 228-240) are too general, and are not responding specifically to this issue.

There are several eye tracking variables that the Toobi system can measure

P 13, l 290: The meaning of this sentence is unclear (e.g., which type of statistical analysis, what results exactly).

P 13, ls 307-309: I’m afraid, I still do not understand how the ESD is related to approach-avoidance motivation. ESD is static, it should not change during eye-tracking data recording, however the stimuli are expected to induce a dinamic change in participants affective responses (Food vs. Neutral). (The Toobi system can offer several metrics such as interval metrics, event metrics, AOI fixation, AOI visit and many more).

RESULTS

More explanation is needed about how the age was treated as a variable in LMMs.

In sum, I do not suggest the manuscript for acceptance in its current form due to some serious conceptual and methodological concerns (especially regarding the eye-tracking).

Reviewer #2: Thank you for the opportunity to review this manuscript, which deals with a very interesting topic. By means of an experimental lab study visual food stimuli (Haute Cuisine vs. local food) were compared to non-food related stimuli with regard to emotional responses on different levels (subjective responses using SAM, physiological responses using EDA, approach-avoidance behavior using ESD). Age was also investigated as moderator variable. A key strength of the manuscript is that multiple levels of emotion were measured to provide a comprehensive picture of subjects' emotional responses. But I have several concerns with the manuscript in the current form, which I will summarize below.

Major concern:

Empirical approach:

- My major concern relates to the internal validity of the experimental setup. The study aims to compare two different kinds of food-related stimuli with non-food stimuli. The HC stimuli are supposed to be ”well-designed,complex and attractive”, whereas local food is described as “careless design, simple and unattractive” (L149/50). As is also well illustrated in the discussion, stimuli differ according to other aspects than HC vs. “local food” that have not been controlled (e. g. L405- emotional valence of the stimuli was not controlled). Differences between the two categories of food stimuli were interpreted in terms of aesthetics (L431), but there might be other explanations for differences or a lack of differences Individual differences in preferences and prior experience were also not controlled for, which is also mentioned in the discussion (e.g. L476f). Primarily because of this lack of standardization and control of the experimental stimuli and confounding variables, the validity of the results is severely limited and it is questionable whether the title “How much is a chef’s touch worth?” does reflect the used approach. These aspects are well discussed in the limitations section, but unfortunately, they can rarely be addressed in a revised version of the manuscript without setting up a new empirical study.

Other concerns:

- Overall: There seem to be several minor mistakes regarding the English language. Proofreading by a native speaker is recommended.

Introduction:

- Overall: There are several quite heterogenous topics (reactions to food stimuli in general, effects of age, HC vs. other food stimuli, theory of emotion…) within the introduction section that does not use subheadings. The use of subheading might improve the readability of the introduction. For example, after a short introduction on the research gap addressed by the study, the different aspects from theory and former research could be described. The introduction should also deliver a more systematic approach in reporting findings for the different levels of emotions. In particular, approach behavior, which is central to the empirical framework, has so far hardly been addressed in the introduction.

-L55ff: Several lines are used for describing controlled lab settings in contrast to more naturalistic settings. This seems to be relevant for the method section (controlling confounding variables…), but does not bring much input in the introduction section, which aims to develop the research questions based on theory and previous research.

-L131: “impulsive and immature”- can this be concluded based on the research findings?

-L135f: “the research on effect of aesthetics of HC dishes … is scarce in general” – could you report in more detail what is known about HC aesthetics?

-L152ff: There is only a general hint, that hypotheses are „based on previous literature”, but the three hypotheses should be derived from theory/research in more detail. Especially the moderator hypothesis requires more explanation. The entire introduction should be examined to see how the objectives and hypotheses of the paper can be derived in more detail and in a more structured way (see first comment for the introduction section).

Method and Results:

-L168f: “35 Portuguese students” vs. L172f: “most of the sample were students”. Maybe in L168 “Portuguese subjects” are meant?

-L169: Since age is an important variable, more information regarding the distribution of the variable might be helpful and should be included (e. g. histogram, distribution in several age groups, is the oldest person with 70 years an outlier…)

-L181ff: Were the stimuli presented in randomized order / in blocks (e. g. all HC stimuli, then all local food stimuli…)?

-L181ff: More details regarding the experimental food stimuli (HC vs. local) and the criteria for selection should be presented in the methods section. What were the reasons for the selection of the food stimuli used in the study (there might be a huge variety of HC and local food pictures)? What did they show? Was there an attempt to keep certain characteristics constant between the two categories (e. g. colors, number of objects, background, calories of the food…)? Was there some kind of selection process / pretest of the stimuli? Is the presentation of local food stimuli the opposite of “a chef’s touch” (title)? Are there other forms of non-HC food stimuli? What were the reasons for using local food (in contrast to HC)?

-L183: To make the selection of the IAPS pictures more transparent, it is recommended to name the numbers of the used pictures and/or to describe the criteria for selection (topics or other aspects of the pictures…).

L188: SAM is described/mentioned twice- it would be sufficient to describe it in the measures section.

- Based on the discussion section, I assume you did not control for any individual differences (e. g. food and restaurant preferences, experience…)? Are there any additional variables you could use as control variables in reanalyzing the data?

- L264: Were there any reasons why the subjects attached the electrodes themselves? Was there some kind of instruction and did you check the placement?

-L318: Age is described as a factor, but it seems, it was entered as continuous variable (not as a categorical variable). In case that is correct, the term “covariate” would be more appropriate than “factor”.

- The sample size is quite small – especially when interaction effects with age are analyzed. The statistical power is limited and the non-significant effects should be interpreted carefully. Sample size is mentioned in the limitations section, but it should also be discussed in terms of statistical power. Since there are several non-significant results for different dependent variables, the knowledge gained from the study is limited.

Discussion:

- The limitations of the study are well discussed in the discussion and limitations section!

-L431: It is unclear, whether the results reflect aesthetics, since there were no aesthetics items/measures included in the study.

- Since several confounding variables were not controlled and due to the small sample size and non-significant effects, the conclusions derived from the results appear uncertain.

6. PLOS authors have the option to publish the peer review history of their article (what does this mean?). If published, this will include your full peer review and any attached files.

Reviewer #1: No

Reviewer #2: No

---

## [Author Response · Author response to Decision Letter 0]

4 Oct 2023

Journal Requirements:

R: Acknowledged. We amended it accordingly. 

“This study was funded by the (BLINDED FOR REVIEW PURPOSES).”

R: Thank you. We have changed it in the cover letter.

3. We note you have included a table to which you do not refer in the text of your manuscript. Please ensure that you refer to Tables 3 and 4 in your text; if accepted, production will need this reference to link the reader to the Table.

 R: We added now table 3 and 4 in the text.

Reviewer #1: 

In this study the authors investigate subjective and physiological responses to food images and non-food control stimuli. Valence and arousal ratings, electrodermal activity and eye-tracking responses were collected from 35 Portuguese participants. Age-related effects were also examined.

The main advantages of the study are (1) its levels of investigation (subjective and physiological); (2) interesting research question.

Although the paper has several strengths, I do not recommend it for acceptance in its current form. Please, find some critics below.

ABSTRACT

The theoretical background of the study is unclear in the abstract as there is a mismatch between the content in the Background paragraph and in the Conclusions paragraph.

R: The mismatch was corrected (page 1, line 10).

What does the eye-to-screen distance exactly mean (as a variable)? I’m not sure if it is the correct term to use.

R: Acknowledged. In fact, is not a common metric, but the eye-to-screen distance is nothing more than the distance from the eye tracker (screen) to both eyes. It has several designations such as viewing distance or distance to screen. This measure (two data columns for each eye named distance Left and distance Right, respectively) can be accessed through the output of the eye tracker (tobii manual can be found and accessed here

https://www.staff.universiteitleiden.nl/binaries/content/assets/sociale-wetenschappen/faculteitsbureau/solo/research-support-website/equipment/manual-tobii-t120_23122019.pdf

In Tobii Studio analysis software this data can be accessed through a raw data text export function. The distance is given in mm on a straight axis right out from the eye tracker plane. This measure is computed and displayed by every single eye tracking system to ensure that the participant is within the tracking area (usually 44 x22 cm at 70cm), so the both eyes and their movements can be recorded at tracking distance between 50-80 cm. In our study we used this measure in order to quantify approach and avoidance behaviors. Approach and avoidance are amongst the central constructs in psychology—they have been used to understand attitude formation (Van Dessel, Gawronski, Smith, & De Houwer, 2017), social behavior (Strack & Deutsch, 2004) or information processing (Neumann & Strack, 2000). These two motivational orientations are usually defined with reference to the relationship between the organism/body and an object in the external environment (Strack & Deutsch, 2004). Approach (vs. avoidance) is a tendency to decrease (vs. increase) the distance between oneself and the object. According to approach-avoidance theory, the approach behavior indicates a propensity to move toward (or maintain contact with) a desired stimulus. Avoidance indicates a propensity to move away from (or maintain distance from) an undesired stimulus. Motivation is defined as the energization and direction of behavior. The valence of stimuli (pleasantness) is at the core of the distinction between approach and avoidance, with positively valenced stimuli typically leading to approach and negatively valenced stimuli typically leading to avoidance. In our study we quantify leaning forward (approach) versus leaning back or reclining (avoidance) bodily positions (head and torso) which subsequently impacts on eye-screen-distance. Leaning forward will lead to a shorter eye-screen distance, whereas leaning back or reclining (avoidance) leads to larger eye-screen distance. For instance, the measure we used is available in the iMOTIONS software platform (a software platform designed for research in the field of human behavior, neuroscience, and user experience) (https://imotions.com/blog/learning/best-practice/eye-tracking/) which has been used for years in marketing studies. 

INTRODUCTION

The theoretical background is too diverse. The focus point is not clear enough. Why is it important to conduct a multimodal study on affective responses to food images? 

R: Thank you for your comment: We fully reorganized our introduction in order to keep it more focused on the impact of food images on affective/emotional/behavioral responses. Besides, we use now subheadings in order to increase text readability as recommend (pages 3-9). There is a large body of evidence that food images impact our affective/emotional responses because they tap into our sensory experiences, personal memories, and basic human instincts related to our survival (e.g., Bublitz et al., 2019; MacCormack & Lindquist, 2018). As humans are visual-oriented organisms, beautifully presented food can trigger positive emotions because it's aesthetically pleasing (Spence et al., 2015). Vibrant colors, intricate plating, and appealing textures can make us feel pleased and aroused (Zhang et al., 2022). Since food images can trigger emotions, we aimed at examining how different food images (haute cuisine vs. traditional vs. control) can elicit emotional responses and how they are manifested via three dimensions: subjective felling/cognitive, physiological reactions and behavioral reactions. Furthermore, as we age, our perception on food also changes, impacting in our dietary preferences, nutritional choices, and overall eating habits. It is known that elderly individuals may have different emotional associations with food, which might be tied to memories and nostalgia, influencing visual perception and subsequently their preferences (Kremet et al., 2007; Liu et al., 2022). Therefore, it also was our goal to examine if age can act as moderator variable in the relationship between perception of food images and emotional response. We decided to conducted a multimodal study for 3 main reasons:

1) Physiological and behavioral measures are crucial to complement subjective measures (self-report measures), since the subjective measures can be influenced by various factors, including memory bias and social desirability. Therefore, physiological and behavioral measures can provide an objective and independent validation of the reported experiences, allowing more robust results.

2) The combination of physiological, behavioral and subjective measures provides a more comprehensive and accurate understanding of human emotions, as the emotional response is manifested through 3 response levels: cognitions/feelings, physiological changes and actions. We add behavioral our title comprising now all the response levels.

3) Many psychological and physiological processes occur at a non-conscious level and are not accessible through self-report measures and physiological and behavioral measures can reveal these underlying processes. 

What is the gap exactly that the authors fill in with their study?

R: The study explores influence of aesthetics on affective and emotional responses to images of food, confronting two opposing styles of gastronomy: contemporary vs traditional, instead of just using images of Haute cuisine as in previous studies. The aim is to specifically understand whether designer dishes, with careful design of contemporary Portuguese cuisine created by the today´s most famous Portuguese Chef (awarded with two stars in the Michelin Guide), have a greater visual and affective impact on participants of different ages, when compared to theoretically less attractive dishes of traditional Portuguese cuisine where design is not considered in the presentation of the dishes. Images not related to food are also used as a control measure. Usually, these kinds of studies do not combine physiological, behavioral and subjective measures. Our study is thought to provide a more comprehensive and accurate understanding of the affective, emotional, behavioral responses to images depicting Haute Cuisine dishes (in comparison to traditional dishes and control images), examining the potential role of age as moderator variable. 

METHODS

The sample was recruited based on convenience sampling, but the relatively low number of participants and the homogenity of the sample may be a limitation to generalize the results. I strongly encourage the authors to conduct a post hoc sample size calculation and a power analysis.

R : Well-pointed. We used now the sjstasts (Lüdecke, 2022) R package for a priori sample size calculation for a .80 statistical power. Regarding post-hoc power analysis or observed power, current recommendations state that it is inappropriate to compute it after the study has already been done (e.g., Dziak et al., 2020 DOI: 10.1007/s12144-018-0018-1). We add this information in the participants section (page 10, line 228)

Page 8, line 170: The “civil status” of participants should be changed to demographic features, or to use the term sociodemographic variable consequently (see p 11, l 255).

R: We have changed civil status to demographic features.

P 8, ls 168-169: According to the text “ … 35 Portuguese students aged between 18 and 70 years....” . Maybe it should be corrected as the authors earlier mention that “students and faculty” participated in the study (l 166). Did the participants have prior psychological knowledge regarding the current study, or were they ‘naive’?

R: We have changed Portuguese students to Portuguese adults as the sample comprises both students and faculty. We thank the reviewer for detecting this issue. The participants did not have prior knowledge on the study, but they were informed that would be exposed to food images of traditional dishes as described now in page 16 (line 365-366).

It is a question whether the actual motivational state, namely the hunger of participants or specific eating habits or diets could have an effect on their affective responses. Is there any information about the time of last food consumption, and type of it (e.g., main dish, snack, etc.), as well as specific diets and eating habits?

R: More explanations were added (Page 11, line 250)

Please, clarify some details about the stimulus material: (1) Picture ID numbers from the IAPS would be really informative; (2) please, define the difference between the two food pictures or provide an example of each; (3) what was seen on the neutral pictures (e.g., geometric figures, natural scenes, neutral faces?)

R: In the Supplementary materials we included a picture of each stimulus, and the picture ID numbers for the IAPS materials used in our experimental task (https://osf.io/f2tcm). The neutral stimuli were objects retrieved from IAPS.

If I understand it correctly, participants have placed the electrodes to themselves. It could cause measurement error, couldn’t it? Did the authors check it?

R: Well-pointed by the reviewer. The participants have placed the electrodes to themselves but the signal quality was always checked visually on the screen before starting the experimental task. Electrode placement was adjusted by the researcher if the signal was of poor quality. This information was now added at page 15 (line 349).

P 11, l 239: I am afraid, the ESD is not the optimal solution here. It is unclear why (and how) the ESD is informative about approach-avoidant motivation if the head is fixed during eye-tracking registration? The cited papers (lines 228-240) are too general, and are not responding specifically to this issue.

R: thank you for your suggestion. We would agree with the reviewer’s point of view if we had used a forehead/chin rest, which is not our case. We never mentioned in the manuscript the participant’s head was fixed, otherwise ESD wouldn’t be a valid measure to assess avoidance/approach behaviors. Our eye tracking system allows participants to move their head during the experiment and that is why we measured the ESD. The Tobii t60 was developed to work without forehead/chin rest, offering good tracking accuracy and stability even without this additional support. We do know that using a forehead/chin rest chin rest helps to stabilize the participant's head and increases precision, but as we mentioned before, participants were not constrained by a fixed head position, allowing for natural head and torso movements (related to motivated behavior (avoidance -approach) we were interested in. Moreover, we wanted record this posture-related metric as an implicit measure (the participant was not aware of it), so he/she could naturally interact with the presented images, increasing the ecological validity of the experiment. As previously denoted, there are two motivational orientations/behaviors that are usually defined with reference to the relationship between the organism and an external stimulus (in our case food-related images)- the approach/avoidance behaviors. According to approach-avoidance theory, the approach behavior indicates a propensity to move toward (or maintain contact with) a desired stimulus. Avoidance indicates a propensity to move away from (or maintain distance from) an undesired stimulus. High pleasant stimuli typically leading to approach and unpleasant stimuli typically leading to avoidance. In other words, leaning forward to screen (smaller ESD) versus leaning back to screen (larger ESD). 

There are several eye tracking variables that the Toobi system can measure

P 13, l 290: The meaning of this sentence is unclear (e.g., which type of statistical analysis, what results exactly).

R: Amended. The sentence you mentioned was totally redundant with the previous written sentence. Therefore, it was removed from text. 

P 13, ls 307-309: I’m afraid, I still do not understand how the ESD is related to approach-avoidance motivation. ESD is static, it should not change during eye-tracking data recording, however the stimuli are expected to induce a dinamic change in participants affective responses (Food vs. Neutral). (The Toobi system can offer several metrics such as interval metrics, event metrics, AOI fixation, AOI visit and many more).

R: The Tobii eye tracking systems allow us recording extraocular, intraocular, periocular and posture-related movements. As we mentioned before, we had no forehead/chin rest, so the participant’s head and torso could be freely moved (within a given head tracking area, of course). So, the ESD, a head/posture-related metric, was computed based on two data columns for each eye named distanceLeft and distanceRight, respectively. With the ESD we were able to quantified the leaning forward (approach = smaller ESD) versus the leaning back or reclining (avoidance = longer ESD) even for micro/small oscillations. This measure is being used in the iMOTIONS software in marketing studies (https://imotions.com/blog/learning/best-practice/eye-tracking/). 

RESULTS

More explanation is needed about how the age was treated as a variable in LMMs.

R: Acknowledged. We replaced “factors” with “effects”. Age I cannot be a factor since is a continuous variable and it was corrected. Furthermore, we mention now that age was included as a continuous variable in the model and that the potential moderating effect was examined via simple effects. The age moderating in the Linear Mixed models was analyzed and interpreted the same way as for General Linear Models. As we examined Categorical (type of image) X Continuous (age) interactions, we tested the simple effects for age (one standard deviation above, mean, and below the mean) at all levels of the factor, i.e., all type of images (haute cuisine vs. traditional vs control). We added now this information (pages 18, line 406 - 410)

In sum, I do not suggest the manuscript for acceptance in its current form due to some serious conceptual and methodological concerns (especially regarding the eye-tracking).

R: We now hope to have clarified the use of the ESD, how is computed and its validity for quantifying approach and avoidance behaviors.

Reviewer #2: 

Thank you for the opportunity to review this manuscript, which deals with a very interesting topic. By means of an experimental lab study visual food stimuli (Haute Cuisine vs. local food) were compared to non-food related stimuli with regard to emotional responses on different levels (subjective responses using SAM, physiological responses using EDA, approach-avoidance behavior using ESD). Age was also investigated as moderator variable. A key strength of the manuscript is that multiple levels of emotion were measured to provide a comprehensive picture of subjects' emotional responses. But I have several concerns with the manuscript in the current form, which I will summarize below.

Major concern:

Empirical approach:

- My major concern relates to the internal validity of the experimental setup. The study aims to compare two different kinds of food-related stimuli with non-food stimuli. The HC stimuli are supposed to be ”well-designed,complex and attractive”, whereas local food is described as “careless design, simple and unattractive” (L149/50). As is also well illustrated in the discussion, stimuli differ according to other aspects than HC vs. “local food” that have not been controlled (e. g. L405- emotional valence of the stimuli was not controlled). Differences between the two categories of food stimuli were interpreted in terms of aesthetics (L431), but there might be other explanations for differences or a lack of differences Individual differences in preferences and prior experience were also not controlled for, which is also mentioned in the discussion (e.g. L476f). Primarily because of this lack of standardization and control of the experimental stimuli and confounding variables, the validity of the results is severely limited and it is questionable whether the title “How much is a chef’s touch worth?” does reflect the used approach. These aspects are well discussed in the limitations section, but unfortunately, they can rarely be addressed in a revised version of the manuscript without setting up a new empirical study.

R: The stimuli were standardized: images of signature dishes, where each dish is in fact designed to cause a visual impact and live up to the maxim that “the eyes eat first”, versus typical dishes of popular Portuguese recipes, where primacy is given taste rather than aesthetics. It should be noted that: i) all participants were Portuguese and ii) they were all familiar with traditional Portuguese cuisine. In this sense, the novelty element would theoretically be the images of Chef Avillez's dishes, because they are exclusive and not massified. If, on the one hand, each individual has their own taste and food specificities in terms of diet, on the other hand, we consider that for the purposes of the study there is a cultural and historical component, that is, “we are what we eat” (Hall & Mitchell, 2000), which is common to a people, which is called the gastronomic identity matrix. However, the results show that the stimuli provoked by Haute cuisine dishes are more pronounced in older participants, possibly because this type of cuisine is interpreted as an art form (Madeira et al., 2021) and in this sense, the maturity of the older participants is, in our opinion, decisive because they understand that the dish created by the Chef is much more than just food. In the opposite direction, younger participants are more stimulated by images of high-calorie dishes of traditional Portuguese cuisine. This can be explained, on the one hand, by the fact that young people are attracted to more caloric foods and, on the other hand, by the fact that new generations are gradually moving away from traditional gastronomy, which has been neglected by what has been called “ junk food”, that includes hamburgers, pizzas, etc. In other words, traditional dishes are an element of novelty for younger people, because they are not consumed as often as they were by the previous generations.

Other concerns:

- Overall: There seem to be several minor mistakes regarding the English language. Proofreading by a native speaker is recommended.

R: The text was revised again by a native English speaker.

Introduction:

- Overall: There are several quite heterogenous topics (reactions to food stimuli in general, effects of age, HC vs. other food stimuli, theory of emotion…) within the introduction section that does not use subheadings. The use of subheading might improve the readability of the introduction. For example, after a short introduction on the research gap addressed by the study, the different aspects from theory and former research could be described. The introduction should also deliver a more systematic approach in reporting findings for the different levels of emotions. In particular, approach behavior, which is central to the empirical framework, has so far hardly been addressed in the introduction.

R: Thank you for pointed that out. We excluded some introductory sentences and we fully reorganized our introduction via subheadings as recommend (page 3 to 9) 

-L55ff: Several lines are used for describing controlled lab settings in contrast to more naturalistic settings. This seems to be relevant for the method section (controlling confounding variables…), but does not bring much input in the introduction section, which aims to develop the research questions based on theory and previous research.

R: Thank you for your suggestion. We highlighted this information in the methods when describing the experimental task.

-L131: “impulsive and immature”- can this be concluded based on the research findings?

R: Thank you for your suggestion. We highlighted this information in the methods when describing the experimental task.

-L135f: “the research on effect of aesthetics of HC dishes … is scarce in general” – could you report in more detail what is known about HC aesthetics?

R: Explanations were added (page 7-8, line 149 to 156).

-L152ff: There is only a general hint, that hypotheses are „based on previous literature”, but the three hypotheses should be derived from theory/research in more detail. Especially the moderator hypothesis requires more explanation. The entire introduction should be examined to see how the objectives and hypotheses of the paper can be derived in more detail and in a more structured way (see first comment for the introduction section).

R: Explanations were added (page 7-8, line 149 to 156).

Method and Results:

-L168f: “35 Portuguese students” vs. L172f: “most of the sample were students”. Maybe in L168 “Portuguese subjects” are meant?

R: Thank you for detecting this issue. We have corrected to “Portuguese adults”.

-L169: Since age is an important variable, more information regarding the distribution of the variable might be helpful and should be included (e. g. histogram, distribution in several age groups, is the oldest person with 70 years an outlier…)

R. Thank you for the suggestion. We present now a histogram (Figure 4) showing the age distribution of study participants (N = 35). We also presented the values for skewness and kurtosis. We also referred that whenever an observation was above + 3SD or below – 3SD was considered an outlier. Based on this rule, no outliers were identified and removed. (page 19, line 429 – 438)

-L181ff: Were the stimuli presented in randomized order / in blocks (e. g. all HC stimuli, then all local food stimuli…)?

R: well-pointed. Stimuli were randomly presented. We add now this information at page 12, line 270.

-L181ff: More details regarding the experimental food stimuli (HC vs. local) and the criteria for selection should be presented in the methods section. What were the reasons for the selection of the food stimuli used in the study (there might be a huge variety of HC and local food pictures)? What did they show? Was there an attempt to keep certain characteristics constant between the two categories (e. g. colors, number of objects, background, calories of the food…)? Was there some kind of selection process / pretest of the stimuli? Is the presentation of local food stimuli the opposite of “a chef’s touch” (title)? Are there other forms of non-HC food stimuli? What were the reasons for using local food (in contrast to HC)?

R: More details were added in the methods section (page 11, line 250). Regarding the last question, yes, the authors think that the images of traditional Portuguese food are completely antagonistic to those of HC, for the reasons already explained above and which go through the harmony that chefs seek between aesthetics and taste, that is, there is the concern to surprise customers/consumers first with the visual appearance of the dishes.

-L183: To make the selection of the IAPS pictures more transparent, it is recommended to name the numbers of the used pictures and/or to describe the criteria for selection (topics or other aspects of the pictures…).

R: We have prepared a supplementary file that contains the images and IDs of the IAPS stimuli used in our study (https://osf.io/f2tcm). The neutral images depicted objects without emotional valence.

L188: SAM is described/mentioned twice- it would be sufficient to describe it in the measures section.

R: thank you for pointing out this redundancy. We have changed it

- Based on the discussion section, I assume you did not control for any individual differences (e. g. food and restaurant preferences, experience…)? Are there any additional variables you could use as control variables in reanalyzing the data?

R: Participants were asked about their diets, namely whether they ate animal protein, and some vegetarians/vegans were excluded from the study, considering the choice of images of dishes from both gastronomic styles. Thus, the hypothesis of a possible rejection of images of dishes of animal origin was excluded. In this sense, all participants had in common the fact that they were culturally familiar with the matrix that defines Portuguese cuisine, as part of the Mediterranean diet. Even in the case of Chef Aviles’s dishes, the cuisine presented is defined as contemporary Portuguese, where most of the dishes are based on traditional recipes, which are reinvented/redesigned. In a future study, we can use other control variables, but considering the objectives defined for this study, we believe that the selection based on the criteria described above is sufficient. 

- L264: Were there any reasons why the subjects attached the electrodes themselves? Was there some kind of instruction and did you check the placement?

R: Well-pointed by the reviewer. The participants placed the electrodes to themselves because it an easy task when they are well instructed while the researcher as setting up the eye tracking system. However, the signal quality was always checked visually on the screen before starting the experimental task. Electrode placement was adjusted by the researcher if the signal was of poor quality. This information was now added at page 15 (line 349).

-L318: Age is described as a factor, but it seems, it was entered as continuous variable (not as a categorical variable). In case that is correct, the term “covariate” would be more appropriate than “factor”.

R: Amended. We replaced “factors” with “effects”. Indeed, Age I cannot be a factor since is a continuous variable and it was corrected. Furthermore, we mention now that age was included as a continuous variable in the model and that the potential moderating effect was examined via simple effects (page 18, line 406 - 410)

 - The sample size is quite small – especially when interaction effects with age are analyzed. The statistical power is limited and the non-significant effects should be interpreted carefully. Sample size is mentioned in the limitations section, but it should also be discussed in terms of statistical power. Since there are several non-significant results for different dependent variables, the knowledge gained from the study is limited.

R: We used now the sjstasts (Lüdecke, 2022) R package for a priori sample size calculation for a .80 statistical power. We add this information in the participants section. 

Discussion:

- The limitations of the study are well discussed in the discussion and limitations section!

R: Thank you.

-L431: It is unclear, whether the results reflect aesthetics, since there were no aesthetics items/measures included in the study.

R: The results reflect the impact of stimuli provoked by two antagonistic approaches to gastronomic aesthetics: Hault Cuisine vs. traditional cuisine. If, on the one hand, in the case of the aesthetics of Haute cuisine, an analogy can be made with the exclusive Chef dishes as paintings, on the other hand, there is no aesthetic sense based on a single set of conventions as in an art form (Madeira et al., 2021), which means that a theoretical ugly dish can also be aesthetically appealing.

- Since several confounding variables were not controlled and due to the small sample size and non-significant effects, the conclusions derived from the results appear uncertain.

R: Thank you for pointed that out. We do agree that gathering a large number of participants is always the best and recommended scenario for any research. Replications cannot replace the added population representativeness that a participant brings, but it is possible to increase statistical power by replicating a measure multiple times in each condition (we presented 18 times for each type of food-related image), but we do not mean to say that participants can be traded for trials. (Goulet & Cousineau, 2019). Therefore, as we had a mixed design, the interaction effect (a crossed effect = within measurement x between measurement) was the most demanding effect in terms of statistical power. This reduced power comes from the fact that a product variable has a reliability equal to the product of the reliabilities of the two variables (Aiken & West, 1991) and the tendency for the product variable to have a nonnormal distribution (Shieh, 2009). We presented now the a priori sample size calculation for a power of .80.

---

## [Editor Report · Decision Letter 1]

9 Oct 2023

How much is a Chef’s touch worth? Affective, emotional and behavioural responses to food images: a multimodal study

PONE-D-23-10020R1

Dear Dr. Pedro J. Rosa,

We’re pleased to inform you that your manuscript has been judged scientifically suitable for publication and will be formally accepted for publication once it meets all outstanding technical requirements.

Kind regards,

Alhamzah F. Abbas, PhD

Academic Editor

PLOS ONE
---

## [Editor Report · Acceptance letter]

18 Oct 2023

PONE-D-23-10020R1 

How much is a Chef’s touch worth? Affective, emotional and behavioural responses to food images: a multimodal study 

Dear Dr. Rosa:

I'm pleased to inform you that your manuscript has been deemed suitable for publication in PLOS ONE. Congratulations! Your manuscript is now with our production department. 

Kind regards, 

on behalf of

Dr. Alhamzah F. Abbas 

Academic Editor

PLOS ONE